# A unified 3D geological model for Germany and adjacent areas

Steffen Ahlers[1], Andreas Henk[1]

[1]Technical University of Darmstadt, FB 11, Institute of Applied Geosciences, Engineering Geology, Darmstadt, 64287, Germany

*Correspondence to*: Steffen Ahlers (ahlers@geo.tu-darmstadt.de)

**Abstract**

3D geological models are an essential source of information for research as well as for the safe and efficient use of the underground. They provide not only a visualization of the subsurface structures but also serve as geometry input for geophysical and numerical models, e.g., gravimetric, mechanical or thermal models. The set-up of a geological model for a
numerical simulation is often a time-consuming task. During the last two decades, several 3D geological models have been created for specific regions in Germany. Up to now only one attempt has been made to combine several of them to a Germany-wide model. However, there are many new models that have not been integrated into this model. Therefore, we present a new Germany-wide 3D geological model combining information of 27 individual models. The model has a resolution of 1 x 1 km$^2$ and is vertically and horizontally subdivided into 146 units. Where possible, the model is extended to neighboring countries,
e.g., the Netherlands, Belgium, France, Switzerland and Austria. In order to combine all models with their different sizes, resolutions and stratigraphic subdivisions, we used a point set approach, which has a number of advantages with regards to its flexibility and usability. To demonstrate the usability, the set-up of a finite-element model is shown as a possible application.

## 1        Introduction

3D subsurface models showing geological units are fundamental for research as well as various applications and are essential
for any safe and efficient use of the underground. Such structural models help not only to visualize the often complex geology but also provide the input geometry for numerical models, e.g., thermal, hydraulic or geomechanical models (Ahlers et al., 2021; 2022a; Anikiev et al., 2019; Arfai and Lutz, 2018; Balling et al., 2013; Koltzer et al., 2022). Such numerical simulations can be used to predict the natural temperature, pore pressure or stress state, as well as how subsurface operations would potentially disturb them. Thus, 3D subsurface models are indispensable when it comes to the assessment of the geothermal
potential of a region, the minimization of induced seismicity or the search for a high-level nuclear waste repository and its long-term safety, to name just a few of the wide range of possible applications.

3D subsurface models can have very different scales ranging from meters to hundreds, or even thousands, of kilometers. In the following, we focus on the scale of Germany and how various, mainly regional models can be combined. Regional 3D models exist for several individual federal states, e.g., North Rhine-Westphalia (Geologischer Dienst NRW, 2022), Hesse (Bär

et al., 2021), Baden-Württemberg (Rupf and Nitsch, 2008) as well as across several federal states or including neighboring countries, e.g., TUNB (BGR et al., 2022), GeoMol (GeoMol Team, 2015a), GeORG (GeORG-Projektteam, 2013), Erzgebirge model (Kirsch et al., 2017). In addition, models for larger regions, e.g., of the North Alpine Foreland Basin (Przybycin et al., 2015), of the Central European Basin System (Maystrenko and Scheck-Wenderoth, 2013) and of the Upper Rhine Graben region (Freymark et al., 2017) exist and a Germany-wide model that combines these three models by Anikiev et al. (2019).

However, a 3D structural model that combines all models - of a regional-scale - for Germany and neighboring countries, such as the Netherlands, Belgium, Switzerland and Austria, is missing. The challenge of setting up such a model lies in integrating the different models with respect to resolution, depth of horizons and stratigraphic subdivisions.

In the following, we will briefly introduce the existing models we combined to form a unified 3D subsurface model of Germany, including some neighboring countries (Ahlers, 2026). The correlations made and additional raw data used are

outlined, but are also documented in detail for each model surface in the supplement. The unified model can be used to extract further geological information, like depth and thickness maps or to generate individual 3D (sub)models for any region desired. In addition, a workflow is shown which allows to create arbitrary finite-element models based on the unified model. Such discretized models can then be parameterized accordingly and used for thermal, hydraulic, mechanical or coupled simulations.

## 2    Model set-up

### 2.1    Data base

We used 27 individual models of different sizes and stratigraphic resolutions to set up a unified 3D structural model for Germany (Fig. 1). In the following, we use short names and individual numbers (shown in bold) for the integrated models. The original names and references are listed in Table S3. The same short names and numbers are used by Ahlers (2026). As we set up this unified model for the prediction of the recent crustal stress state of Germany by geomechanical-numerical

modeling, it covers the same area as the models of Ahlers et al. (2021; 2022a). Almost all surfaces defined in each of the 27 input models have been used for the unified model, with a few exceptions, e.g., tectonic units of the Erzgebirge model (Kirsch et al., 2017; **13**), whose stratigraphic correlation with other horizons is difficult. The succession between the Earth's surface and the top of the crystalline basement is subdivided into 3 to 24 units, depending on the region and the corresponding input models in place. A special case is the integration of the relatively small ($70 \times 50$ km$^2$) Ingolstadt model (Ringseis et al., 2020b;

**17**) with a highly resolved stratigraphy, with 23 units. We included this data set to prove the possibility of integrating models of different scales and resolutions into one single model and to show benefits and limitations of the chosen point set approach (Sect. 2.3). If not already contained in the input models, we created the top of the crystalline basement - an important boundary for many types of numerical simulations - as a surface with additional data, e.g., well data, seismic sections or other geophysical data. Since the input models usually do not integrate units below the basement, we also created the top of the lower crust and

the Mohorovičić discontinuity (Moho) using additional data. The final model has a lateral resolution of $1 \times 1$ km$^2$, which is a compromise between information loss from high-resolution models and a suitable resolution for a large-scale model. The same

lateral resolution has been previously used for gravity and thermal modeling by Freymark et al. (2017) and Anikiev et al. (2019) for their Germany-wide model.

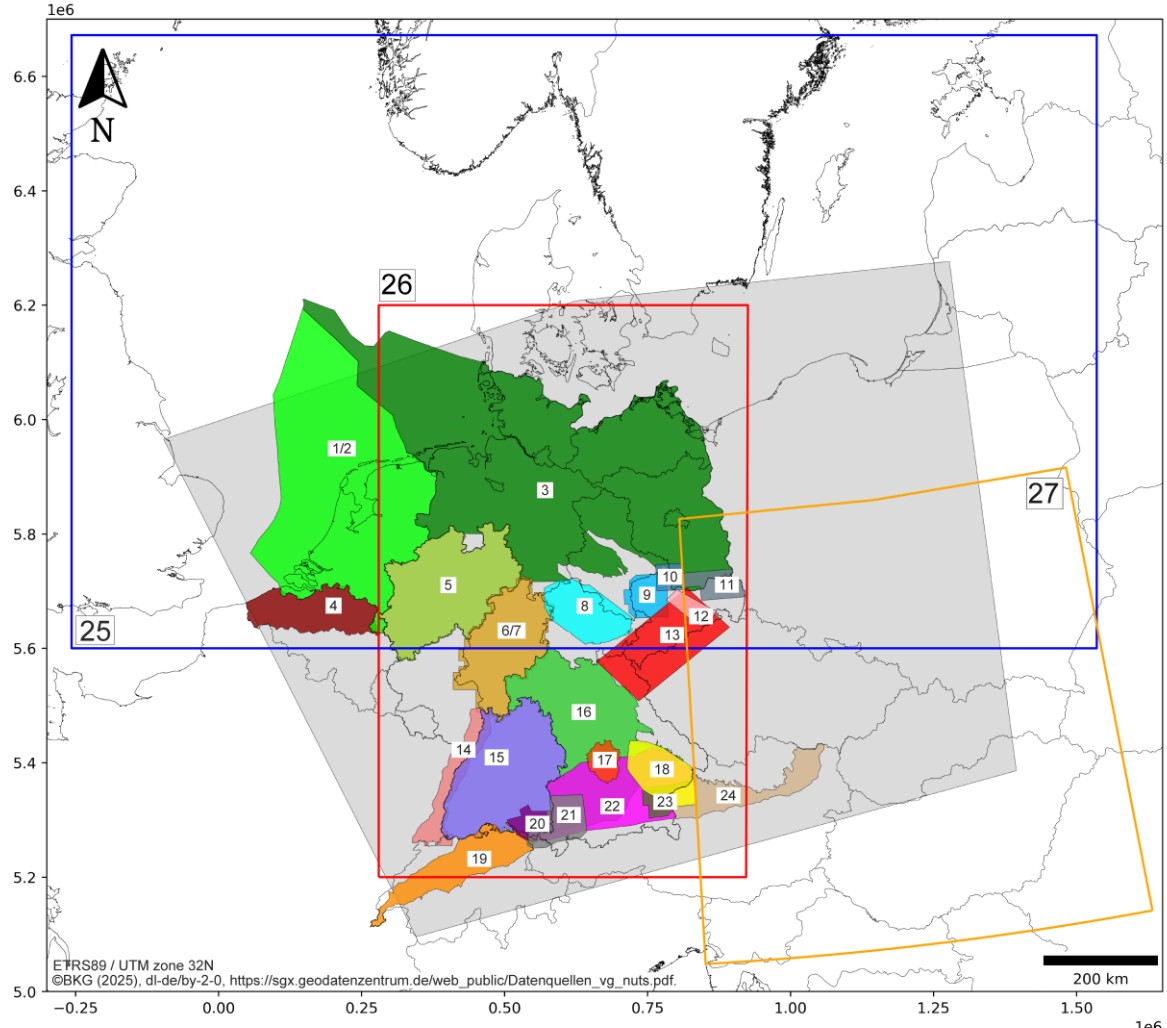

Figure 1: Overview of model area and 3D geological models used: Grey area: Model area. 1/2: Netherlands (TNO, 2019a; 2019b), 3: TUNB (BGR et al., 2022), 4: Vlaanderen (Deckers et al., 2019), 5: Landesmodell NRW (Geologischer Dienst NRW, 2022), 6/7: Hessen (Bär et al., 2021; Weinert et al., 2022), 8: Thueringer Becken (TLUBN, 2014), 9: NW-Sachsen (Görne, 2011), 10: SN Zwischengebiet (Görne, 2012b), 11: Niederlausitz (Görne and Geißler, 2015), 12: Elbtalzone (Görne, 2012a), 13: Erzgebirge (Kirsch et al., 2017), 14: GeORG (GeORG-Projektteam, 2013), 15: Landesmodell BW (Rupf and Nitsch, 2008), 16: Geothermieatlas BY (LfU, 2022), 17: Ingolstadt (Ringseis et al., 2020b), 18: Niederbayern (Donner, 2020b), 19: GeoMol Swiss (Swisstopo, 2019), 20: GeoMol LCA BW (GeoMol LCA-Projectteam, 2015a), 21: GeoMol LCA BY (GeoMol LCA-Projectteam, 2015b), 22: GeoMol FWM BY (GeoMol Team, 2015b), 23: GeoMol UA-UB BY (GeoMol UA-UB-Projectteam, 2015), 24: GeoMol Austria (Pfleiderer et al., 2016), 25: CEBS (Maystrenko and Scheck-Wenderoth, 2013), 26: 3DD (Anikiev et al., 2019), 27: LSCE (Tašárová et al., 2016); The original model names are listed in Table S3; here, short names are used. Coastlines and borders used in this figure are based on the Global Self-consistent Hierarchical High-resolution Geography (GSHHG) of Wessel and Smith (1996).

## 2.2    Model correlation

The first step in creating a unified model that covers an area with a complex geological history (e.g., Plant et al., 2005; McCann, 2008; Meschede and Warr, 2019) is the stratigraphic correlation of all input models. Main challenges are the correlation of models from different countries, e.g., the Netherlands and Germany, from different sedimentary basins, e.g., North German Basin, Upper Rhine Graben and Molasse Basin and from regions with different local stratigraphic terms. Another challenge is the combination of models which are based on different input data, e.g., mainly well-based models like Landesmodell BW (Rupf and Nitsch, 2008; **15**) and mainly seismic-based models like GeORG (GeORG-Projektteam, 2013; **14**). Finally, the variable stratigraphic resolution used in different input models must be considered. Some models provide only the major stratigraphic boundaries, whereas others also provide subunits. An example is shown in Fig. 2. Model A contains four surfaces: top of the Jurassic, top of the Middle Jurassic, top of the Lower Jurassic and base of the Jurassic, whereas model B contains only two of these four surfaces: top and base of the Jurassic. In this example - for an accurate implementation – four units must be considered, e.g., to define proper material properties for a numerical simulation: Upper Jurassic, Middle Jurassic, Lower Jurassic and, in addition, an undifferentiated Jurassic unit.

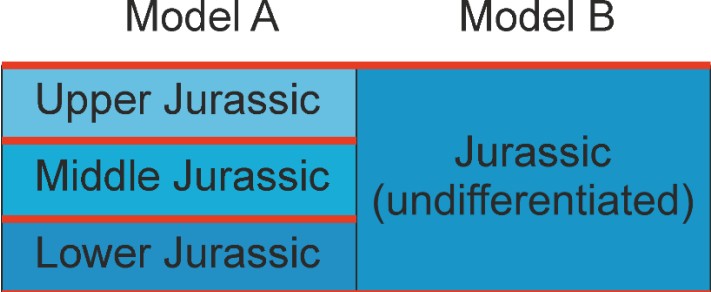

**Figure 2: Sketch illustrating challenge of unit definition of models with different vertical (stratigraphic) resolution. Four units (blue boxes) are defined by four formation interfaces (red lines) of two models. Detailed description see text. (Reiter et al., 2023)**

## 2.3    Point set approach

In order to combine models of different scales, stratigraphic and numerical resolutions and often unknown raw data, we decided not to create new model surfaces. Instead of creating triangulated surfaces, we used a point set approach. The basic modeling concept is shown in Fig. 3. First, a point set with a resolution of 1 x 1 km$^2$ is created. This point set is then projected onto a surface, in this case the topography (Fig. 3a, green line). Next, the projected point set is duplicated, and the duplicated one is projected onto the next underlying surface (Fig. 3b, yellow line). In contrast to the first projection, the projected point set is additionally shifted down by 0.1 m, i.e., a thickness of 0.1 m is applied to the entire lateral extent of the model, even though a unit does not actually exist. This step avoids ambiguous information of different surfaces at a single coordinate, e.g., for a surface pinching-out, like the orange line (Fig. 3b). This step would be not necessary if a surface lies entirely below the overlying surface (yellow and green lines). However, this is not the case for almost all surfaces in our model. We chose a distance of 0.1 m as a compromise between usability during the model set-up and loss of information. A similar distance for

non-existing units, e.g., due to erosional gaps, is used by Anikiev et al. (2019). Considering the 147 surfaces, this minimum distance leads to a shift of up to ~15 m for the lowermost model surface (Moho).

Advantages of the point set model approach are visualized by Fig. 3c-f. If two overlapping surfaces exist (Fig. 3c, purple and pink lines) it is not necessary to cut these or to generate a new surface, which can take several hours per surface depending on size and resolution. In such a case, the projection is staggered (Fig. 3c-d). First, the duplicated and downshifted point set is projected onto the surface with the lowest reliability, in this case the purple one, then the projection is done onto the more reliable surface, in this case the pink one (Fig. 3d). The order of projection is determined according to various criteria, e.g.,

model resolution, amount of raw data or year of model creation. Another advantage of the point set approach is the integration of model surfaces, which occur only locally, e.g., in one single model (Fig. 3e, red line) or to consider the precise definition of stratigraphic units (Fig. 2). Furthermore, if an adapted surface should be integrated (Fig. 3f, dashed orange line), the existing point set can be updated quickly. We used SKUA-GOCAD to apply the point set approach.

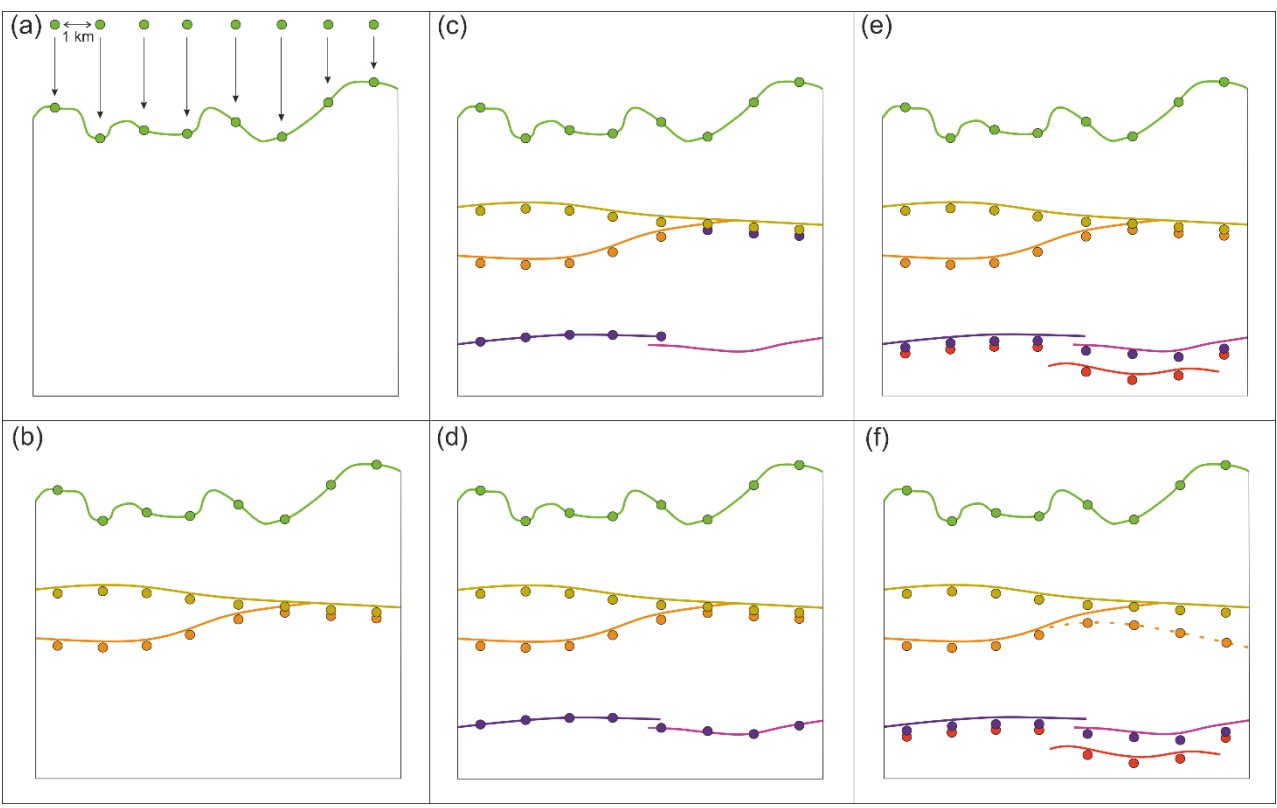

**Figure 3: Sketch of the point set approach used. Details are described in the text. (Reiter et al., 2023)**

# 3 Results

## 3.1 Stratigraphic correlation

Based on the 27 models (Fig. 1), we defined 89 different surfaces. The result of the stratigraphic correlation is summarized in Table S1; a small excerpt is shown in Table 1. Each individual surface is listed in one row and is labeled with its ID and surface name. The IDs are categorized as follows: 00xx stratigraphy independent surfaces, 01xx Quaternary, 02xx Cenozoic, 03xx Cretaceous, 04xx Jurassic, 05xx Triassic, 06xx Permian, 07xx Carboniferous, 08xx Devonian, 09xx Variscan nappes, 10xx top basement. In the columns to the right of the surface name, all models used are listed, i.e., if a model contributes data to a surface, it is documented in the corresponding row. Information is given as follows: original file name – surface name (additional information). The rightmost column "Literature" lists if we used literature for the stratigraphic correlation in addition to the model descriptions and the stratigraphic table of Germany (DSK, 2016).

Table 1: Excerpt from the table "stratigraphic correlation" attached to this paper (Table S1).

| ID | surface name | Netherlands | TUNB | GMTED | … | Literature |
|---|---|---|---|---|---|---|
| 0001 | Topography | | | mn30_grd | … | (Danielson and Gesch, 2011) |
| … | … | | | … | … | … |
| 0222 | Base Selandian | 2_NLNM_tvd_on_offshore_merge_DGM50_ED50_UTM31– Base North Sea Super Group (Top Dan, Basis Selandian) | t - Basis Tertiär (Top Dan, Basis Selandian) | | … | (Doornenbal and Stevenson, 2010) |
| …. | … | … | … | … | … | … |

## 3.2 Model units

Based on the stratigraphic correlation, we defined the final model units. The results are summarized in a second table (Table S2); an excerpt is shown in Table 2. The structure is similar to Table S1 and the excerpt shows the same simple example as shown in Fig. 2. Within the TUNB model (BGR et al., 2022; **3**) the Jurassic is subdivided into three subunits Lias, Dogger, Malm (Lower, Middle and Upper Jurassic), while in the 3DD model (Anikiev et al., 2019; **26**), only one Jurassic unit exists. Therefore, we defined four individual units. In addition to Table S1, we extended the geological categorization of the IDs by: 11xx top crystalline basement, 12xx base upper crust, 13xx base lower crust. Furthermore, we extended categorization of IDs 00xx to 10xx to take into account if several units are defined by one surface (Fig. 2). The final model contains 147 surfaces, i.e., 146 units: 131 sedimentary units, 8 upper crustal units and 7 lower crustal units.

**Table 2: Excerpt from the table "model units" attached to this paper (Table S2).**

| ID | unit name | … | TUNB | | 3DD | … |
|------|-----------|-----|-----------|--|---------------------|-----|
| … | … | … | … | | … | … |
| 0404 | Malm | … | 05_ST_jo | | | … |
| 0413 | Dogger | … | 06_jm | | | … |
| 0416 | Lias | … | 07_ju | | | … |
| 0418 | Jurassic | … | … | | 10_Mesozoic_Triassic | … |
| … | … | … | … | | … | … |

## 3.3 Presentation of results

In addition to the point data sets Ahlers (2026) provides a plot for each of the 146 model units. Furthermore, 11 plots of combined model units are presented: Cenozoic, Cretaceous, Jurassic, Triassic, Zechstein, Rotliegend, PrePerm, Carboniferous, Devonian, upper crust, lower crust. As an example, the combined plot of the Cenozoic model units (0102-0106, 0201-0244, 0307 and 0417) is shown in Fig. 4. We chose this example since 20 of the 27 models used contribute to these Cenozoic units.

In general, all plots are divided into four subfigures. The upper left subfigure shows the depth of the unit base and the lower left the thickness of the unit, which is also displayed as a histogram in the lower right subfigure. In addition, the total area of the unit extent is given above the histogram. The upper right subfigure shows the input data, color-coded according to the input models. The references of models used are displayed to the right of the histogram, in the same order as in the legend of the upper right subfigure. The reference numbers are similar to those in Fig. 1 and Table S3. To account for outliers in the plots,

the most extreme 1% of depth and thickness values are not considered for the color bars. The entire model area is indicated by a red rectangle. The hatched area indicates parts without high stratigraphic resolution models (Fig 1.; **1**-**24**).

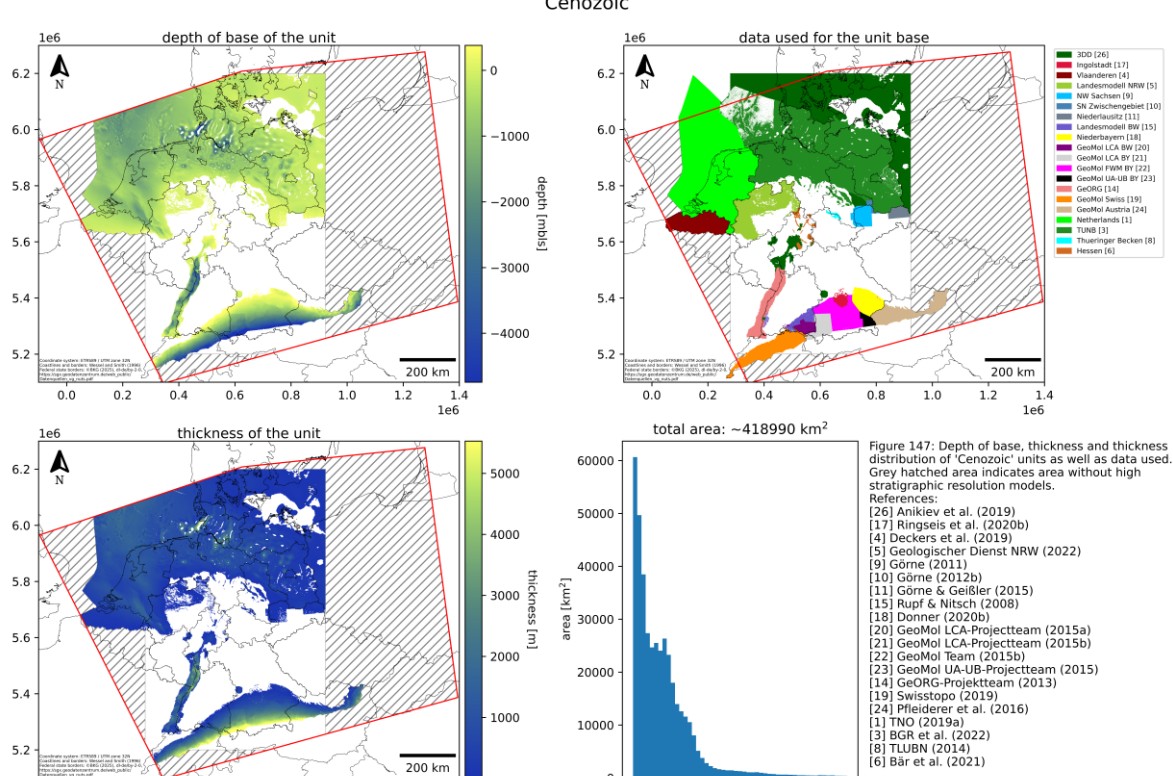

**Figure 4: Depth of base (upper left), thickness (lower left) and thickness distribution (lower right) of combined Cenozoic units (ids: 0102-0106, 0201-0244, 0307, 0417) as well as data used (upper right) as an example of the figures published with the unified model (Ahlers, 2026). A detailed description is given in the text. Coastlines and borders used in this figure are based on the Global Self-consistent Hierarchical High-resolution Geography (GSHHG) of Wessel and Smith (1996).**

Some plots by Ahlers (2026) show minor differences between the "depth" and "thickness" subfigures and the "data used" subfigure. Since the resolution of input models - shown in the "data used" subfigure - sometimes differs from the 1x1 km² model resolution in the other subfigures. For example, unit 0227: the input data of the TUNB model (BGR et al., 2022; **3**) in the North Sea has a very low resolution; therefore, gaps seem to appear in the model. We did not adjust these since this would imply a higher resolution of the input data. Another example of minor deviations between the subfigures is indicated for unit 0237, where the area of the "data used" is larger than the depth and thickness of unit 0237. This is due, on the one hand, to thicknesses that are very low and not resolved in the model (less than 4.4 m) and, on the other hand, to extents that are very small and not covered by the 1x1 km² grid. Deviations occur between the "data used" subfigure and the "thickness" and "depth" subfigures for several units in northern Bavaria, e.g., 0416. Since a 3D geological model is unavailable for this area, we used isolines from the Geothermieatlas BY (LfU, 2022; **16**). In addition, results of a 2D seismic campaign by Fazlikhani et al. (2022) were used. In the vicinity of this seismic survey we generated surfaces using additional data from deep boreholes located further south (Reinhold, 2005). We did not use data from the Geothermieatlas BY (LfU, 2022; **16**) for these surfaces

because the raw data is unavailable. Consequently, some units, e.g., 0526, exhibit substantial differences between the seismic data and the Geothermieatlas BY (LfU, 2022; **16**).

The Zechstein consists of two units (0601, 0602), a salinar and a non-salinar. Since only the 3DD model (Anikiev et al., 2019; **26**) distinguishes between these, we extended this subdivision using Grabert (1998), Seidel (2003), LGB-RLP (2005), Wong et al. (2007), Bachmann et al. (2008), Reinhold et al. (2014), DSK et al. (2020) and Becker et al. (2021). The so called PrePerm units (1103-1108) are "gap fillers" between the top crystalline basement and the deepest units resolved in the models. This unit is therefore sometimes only found as fragments (1107, 1108) and in some cases it is probably only a modeling relict and can therefore be equated with the crystalline basement, e.g., for the Mid German Crystalline High (1105). Unit 1301 shows large areas with small thicknesses, which are also modeling relicts and show the deviation between original data of the 3DD model (Anikiev et al., 2019, **26**) used for surface 1301 and the surface top upper crust (1203-1208) created for the entire model area. Since the resolution of the top of the upper crust is lower than the original 3DD model (Anikiev et al., 2019, **26**), these relicts occur; similar effects occur for unit 1307.

## 3.4 Generation of a discretized model

The following example shows how a discretized 3D model - ready for parameterization - can be created from the unified model outlined above. For this workflow ApplePY v1.3 (Ziegler et al., 2020b), a tool automating the process of discretization (Fig. 5), is used. In addition to a mesh (Fig. 5a) a structural model provided as point data set (Fig. 5b) is required. The creation of the mesh is not restricted to any specific software, however, it must be provided as an Abaqus *.inp file. An Abaqus *.inp file is a structured text file and a common output format. The mesh and structural model are combined in ApplePY (Fig. 5c), i.e., ApplePY assigns each element to a model unit based on the geological information provided by the point data set (Fig. 5d). A detailed description of this tool and the necessary input files is given by Ziegler et al. (2020a).

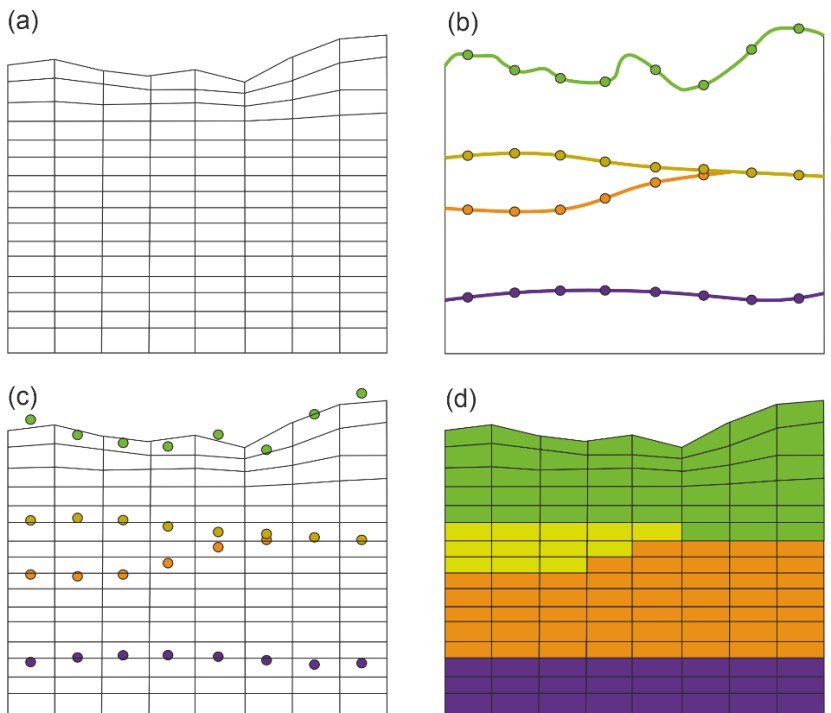

**Figure 5: Sketch of the ApplePY approach (Ziegler et al., 2020b) based on Ziegler et al. (2020a) combining a mesh (a) with a structural model provided as point set data (b-c) to define model units (d). Details are described in the text.**

### 3.4.1 Worked example

To illustrate the working steps, we chose a region of 200 x 200 km$^2$ covering parts of Belgium, Germany and the Netherlands. In this region, five different 3D models have to be considered (TNO, 2019a, 2019b; Deckers et al., 2019; BGR et al., 2022;
Geologischer Dienst NRW, 2022; Fig. 1, **1-5**). The coordinates (in ETRS89 UTM32N) of the area are: y (min) = 5650000, y (max) = 5850000, x (min) = 200000, x (max) = 400000, z (min) = -20000 m, z (max) = 1000 m. The resolution of the mesh is 100 x 100 x 50 elements, whereby the element thickness increases with depth. Increasing element thicknesses are typical for numerical models, e.g., used by Ahlers et al. (2022a). To choose reasonable layers, the stratigraphic correlation in Table S1 should first be considered. In our example, we chose base Cenozoic (in this region, defined as base Zealandian), base
Cretaceous, base Jurassic and top crystalline crust. Once the desired surfaces have been selected, the respective data sets containing the lowest relevant data can be selected in Table S2. In case of base Jurassic, "gg_j_b" from Landesmodell NRW is included in data set 0416, the same applies to Vlaanderen. The corresponding data from Netherlands and TUNB are included in data sets 0413 and 0414, i.e., data set 0416 contains the base Jurassic of all relevant models in the region and is used accordingly as base Jurassic. An overview of all data sets used in the example is given in Table 3.


**Table 3: Overview of surfaces and corresponding data sets used in the example.**

| Surface | Data set |
|---|---|
| Topography | Ahlers_2026_surface_id_0001 |
| Base Cenozoic, i.e., base Seelandian | Ahlers_2026_surface_id_0236 |
| Base Cretaceous | Ahlers_2026_surface_id_0303 |
| Base Jurassic | Ahlers_2026_surface_id_0416 |
| Top crystalline crust | Ahlers_2026_surface_id_1103 |

Once the relevant data sets have been selected, ApplePY (Ziegler et al., 2020b) can be used. Add the chosen data sets (Table 3) to the ApplePY folder, open "create_horizon_file.py" and add the file names to line 12:


*Line 12*

*files=['Ahlers_2026_surface_id_0001.txt','Ahlers_2026_surface_id_0236.txt','Ahlers_2026_surface_id_0303.txt','Ahlers_20 26_surface_id_0416.txt', 'Ahlers_2026_surface_id_1103.txt']*

In addition, adjust the separator to ';' by editing line 28 and 57 from:

*Line 28/57      line = str.split(line)*

to:

*Line 28/57      line = str.split(line,';')*


Then, run "create_horizon_file.py" and open "apple.py". Add the name of the mesh in line 12, define the name of the output file of "create_horizon_file.py" in line 13 and add the unit names in line 14.

*Line 12         geometry = 'example.inp'*

*Line 13         horizons = ['horizons.txt']*

*Line 14         strata = ['Relicts','Cenozoic','Cretaceous','Jurassic','PreJurassic','Crystalline crust']*

"Relicts" is a model unit which occurs due to differences between the topography, i.e., data set 0001 and the surface of the mesh used. Such relict elements occur when the elements defined in the mesh are located above the topography defined by

0001. Therefore, it is possible to create a mesh without topography and remove the relics later, as we did in the example. Next, run "apple.py". The final model is shown in Fig. 6.

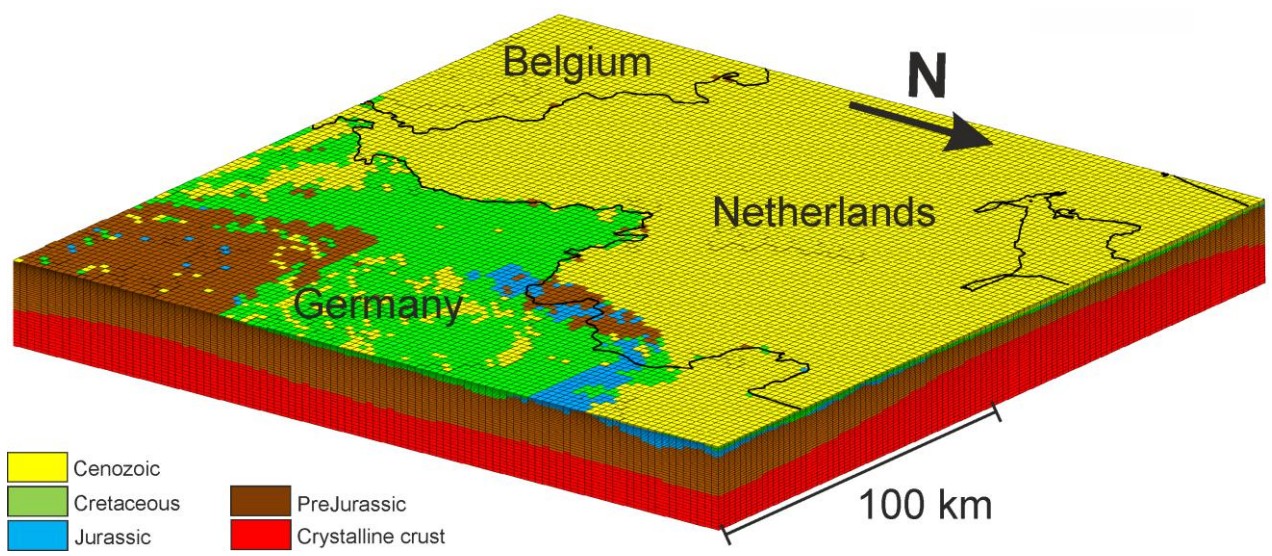

**Figure 6: Discretized model created with data sets of Ahlers (2026) and ApplePY (Ziegler et al., 2020b) visualized with Tecplot. The uppermost unit "Relicts" is removed. The model area is located at the triangle of countries of Belgium, Germany and the Netherlands: y (min) = 5650000, y (max) = 5850000, x (min) = 200000, x (max) = 400000. The base of the model is at a depth of 20 km. Coastlines and borders used in this figure are based on the Global Self-consistent Hierarchical High-resolution Geography (GSHHG) of Wessel and Smith (1996).**

## 4.    Discussion

Due to the diversity and lack of raw data originally used to define the surfaces in the various input models, as well as the heterogeneous distribution of the input data, we decided to use a point set approach with a lateral resolution of 1 x 1 km². The lateral resolution is a compromise between loss of information, pretending a higher resolution in regions where only low-resolution data is available and the manageability of the final model, particularly regarding computing time and main memory. The largest loss of information occurs in regions with highly variable geology in lateral direction, e.g., areas influenced by volcanic activities, halokinetic structures or faulting. The vertical resolution depends mainly on the input data, however, since for technical reasons (Sect. 2) each model unit has a thickness of at least 0.1 m, an error of up to 14.6 m can occur on the deepest units. However, given the thicknesses of the units, the uncertainties of the input data, and the size and purpose of the model, these vertical errors and loss of information are acceptable. For example, the geomechanical-numerical model of Ahlers et al. (2022a) has a maximum vertical element resolution of 250 m and a lateral element resolution of 2.5 x 2.5 km².

The choice of a point set approach as publication format has several reasons. In general, point sets are a common publication form of numerical models, e.g., Maystrenko and Scheck-Wenderoth (2013), Anikiev et al. (2019), Deckers et al. (2019) or Ahlers et al. (2022b). No specific software is required to use point data sets and they can be directly used to quickly create discretized models with ApplePY (Ziegler et al., 2020b; Ahlers et al., 2022a), as demonstrated in Sect. 3.4. Furthermore, the point set approach has several advantages in contrast to common geological "surface" models. Overlapping models do not have to be cut and models with different stratigraphic and lateral resolutions can be combined into one model. However, the

integration of rather small and high-resolution models has some limitations, e.g., some surfaces of the Ingolstadt model (Ringseis et al., 2020b; **17**) have a smaller extent than the point set resolution of 1 x 1 km$^2$. A further advantage of the point set approach is the ability to effectively integrate new or updated data into the existing model.

In areas where large-scale models are unavailable, we often used information from several overlapping models. In such cases, we evaluated the models based on criteria such as model resolution, amount of raw data or year of model creation. However, a quantitative assessment is difficult and accordingly, establishing a general workflow. The level of detail in the documentation varies and the models differ greatly in terms of many assessment parameters, such as model resolution and input data. For example, the amount of input data usually varies with depth, as well as for individual surfaces, e.g., more data are available in economically interesting areas (GeoMol LCA-Projektteam, 2015). A quantitative assessment of the uncertainties of the models would require full access to the raw data in order to, e.g., evaluate the distribution of raw data or the deviations between it and the model surfaces. Experience evaluating this study has shown that the year a model is created is often an indicator of its resolution and data basis, compared with other models from the same area. However, this is only an indication, and we did not used it as fixed criterion. It may also be incorrect for other models or regions not considered here. Ultimately, decisions must be made individually for each region and for each model. However, to illustrate our procedure, we will describe the model ranking within the German part of the Northern Alpine Molasse Basin. To clarify the description, we omitted references to specific models and only included model numbers (Fig. 1 and Table S3).

We used seven different models in the German part of the Northern Alpine Molasse Basin (**15, 17, 18, 20, 21, 22** and **23**). The GeoMol models (**20, 21, 22** and **23**) are divided into the framework models (FWM) and the pilot region models. The pilot region models are characterized by a higher resolution compared to the FWM (GeoMol Team, 2015a). Therefore, we ranked the GeoMol FWM BY (**22**) lower than the pilot region models (**20, 21** and **23**). The FWM of Baden-Württemberg was unavailable, however, it largely corresponds to the Landesmodell BW (**15**) (GeoMol Team, 2015a). Ingolstadt (**17**) has a specified horizontal resolution of 100 m and comprises a relatively large number of geological units compared to other models of the region that Ingolstadt is embedded in (Ringseis et al., 2020a). However, it is difficult to distinguish Ingolstadt from the GeoMol pilot region models, since only for GeoMol UA-UB BY (**23**) a horizontal resolution (of 400 m) is given (Sieblitz, 2019). Ultimately, we assessed the Ingolstadt model (**17**) as more reliable than the GeoMol models (**20, 21, 22,** and **23**). Niederbayern (**18**) does not overlap with Ingolstadt (**17**), but with GeoMol FWM BY (**22**) and GeoMol UA-UB BY (**23**). The horizontal model resolutions for both Niederbayern (**18**) and GeoMol UA-UB BY (**23**) are specified as 400 m (Sieblitz, 2019; Donner, 2020a). Since additional drilling data and parts of GeoMol UA-UB BY (**23**) are integrated into Niederbayern (**18**), we rated Niederbayern (**18**) as more reliable than GeoMol UA-UB BY (**23**) as well as GeoMol FWM BY (**22**). In Baden-Württemberg, GeoMol LCA BW (**20**) and Landesmodell BW (**15**) overlap. Since GeoMol LCA BW (**20**) is based - in parts - on Landesmodell BW (**15**), it is significantly more detailed and incorporates new data, we ranked it higher than Landesmodell BW (**15**). Although no comprehensive documentation of GeoMol FWM BY (**22**) is available, GeoMol Team (2015a) indicates that GeoMol FWM BY (**22**) integrates more data than Landesmodell BW (**15**), especially seismic data. Additionally, faults in GeoMol FWM BY (**22**) are implemented with a dip rather than vertically, as in Landesmodell BW (15), which represents the

geology in the vicinity of the faults more precisely. Based on this evaluation, we prioritized the models as follows: 1 Ingolstadt (**17**) and Niederbayern (**18**); 2 GeoMol LCA BW (**20**), GeoMol LCA BY (**21**), GeoMol UA-UB BY (**23**); 3 GeoMol FWM BY (**22**); 4 Landesmodell BW (**15**).

The unified model is based on 27 models (Fig. 1). The federal state models from Baden-Württemberg (**15**), North Rhine-Westphalia (**5**), Hesse (**6**), and Thuringia (**8**) form the basis, along with the TUNB model (**3**) from the northern federal states. We filled the remaining gaps using either larger-scale models (3DD, CEBS and LSCE; **25**, **26** and **27**) or smaller-scale models, e.g., in Bavaria (Ingolstadt, Niederbayern, GeoMol LCA BY, GeoMol FWM BY and GeoMol UA-UB BY; **17**, **18**, **21**, **22** and **23**). If available, we used additional data to fill remaining gaps, e.g., in northern Bavaria (Geothermieatlas BY; **16**). Since the top of the crystalline basement is an important boundary for geomechanical modeling, we devoted more effort to this surface and used a total of 26 additional data sets (Fig. 124-130 in Ahlers, 2026). When selecting or excluding models, we considered both the model area and the integrated model surfaces. Small-scale models covering only a few percent of the total model area (~1000 x 1250 km²) or representing only thin geological layers have been disregarded. One exception is the relatively small (70 x 50 km²) Ingolstadt model (**17**), which has a high geological resolution and is used to test the point set approach. Models comprising tectonically or petrologically defined surfaces have only been used in some cases, e.g., Erzgebirge model (**13**), since correlating tectonic or petrologic surfaces with stratigraphic surfaces is challenging. Our main focus when selecting the models was on the area of Germany. However, where available, we also used data from neighboring countries, such as the Netherlands, Belgium, France, Switzerland or Austria.

However, we did not use a number of models. For example, several models from Bavaria with an area of 10 x 10 km², such as those of Gershofen (Landmeyer, 2019) or Schweinfurt (Schumann, 2015), are not significant for a Germany-wide model. An integration would be suitable if those small models were available across the board. Technically, an integration is possible, as demonstrated by the Ingolstadt model. Saxony is another federal state with numerous available models, but only some of these are used, and often only partially. Many of these models are small-scale and have tectonic or petrological units that are difficult to correlate with stratigraphic units, while others focus on Cenozoic strata (Geißler et al., 2014). Additionally, we did not use certain models because they have been replaced by more recent, updated models. For example, Baldschuhn et al. (2001) and models of several northern federal states have been replaced by the TUNB model (**3**). In summary, many published models that have not yet been integrated could contribute to the unified model. However, the effort required for a Germany-wide model would not be justified by the results, as they often only improve quality locally. For example, thin units cannot be resolved numerically in a Germany-wide finite-element model. Nevertheless, this would be technically possible, as demonstrated by the Ingolstadt model. However, in this case, the horizontal resolution should be increased and the minimum distance between layers should be decreased to less than 0.1 m.

## 5.    Conclusions

Creating a geological model as geometry input for numerical models often takes a significant amount of time, especially if different data sets must be combined. The unified geological model of Germany and adjacent areas of Ahlers (2026) is intended to replace this labor-intensive work step, as far as possible, especially for large-scale models, or at least to simplify this work step by providing the stratigraphic correlation between models and regions. To our knowledge, this model is the most detailed geological model of Germany. It combines 27 models of different sizes from Germany and neighboring countries, including the Netherlands, Belgium, France, Switzerland and Austria. It contains 147 surfaces, i.e., 146 units and is provided as a point data set with a resolution of 1 x 1 $km^2$.

A comprehensive supplement documents the assumptions made. 157 figures of each individual unit and some combined units visualize the results. Due to heterogeneous input models and the overall size of our model, we used a point set approach, i.e., we almost did not create new surfaces. This approach allows to integrate overlapping surfaces without cutting them or model surfaces that only occur locally. It also allows for the quick implementation of new or updated data. Especially if ApplePY (Ziegler et al., 2020b) is used, it is possible to create a discretized 3D finite-element model within a very short time, which can then be parameterized with mechanical, thermal or hydraulic material properties as required. The final model resolution of 1x1 $km^2$ is reasonable for large-scale models, for studies focusing on small-scale structures the original data sets should be used.

Additional small-scale models or models incorporating tectonic or petrological data could improve the unified model. However, for the purpose of this model, a geomechanical model of Germany, it would have required a disproportionately high amount of effort. In order to facilitate the integration of new data and the assessment of overlapping models, which is always, to some extent, subjective, as well as to adjust if a different ranking is preferred, we aim to document the use of input data as accurately as possible. Further improvements could include closing existing gaps, e.g., Rhineland-Palatinate or integrating data from additional countries. Homogenizing the individual model units, e.g., calculating uniform units for the entire model area, would further increase the applicability of the unified model.

**Data availability**:

The model is published as Ahlers (2026) and is available under https://doi.org/10.48328/tudatalib-1791

**Supplement**:

Table S1: Stratigraphic correlation

Table S2: Model units

Table S3: Overview of input models

**Author contribution**:

SA: Conceptualization, data curation, investigation, methodology, validation, visualization, writing (original draft preparation, review and editing).

AH: Conceptualization, funding acquisition, project administration, supervision, validation, writing (original draft preparation, review and editing).

**Competing interests**:

The authors declare that they have no conflict of interest.

**Acknowledgements**

We would like to thank the Geologische Bundesanstalt Wien, the Landesamt für Geologie, Rohstoffe und Bergbau Baden-Württemberg, the Bayerische Landesamt für Umwelt, the Hessische Landesamt für Naturschutz, Umwelt und Geologie, the Geologische Dienst NRW, the Sächsisches Landesamt für Umwelt, Landwirtschaft und Geologie and the Thüringer Landesamt

für Umwelt, Bergbau und Naturschutz for providing 3D geological models and additional data. This research is part of the project SpannEnD 2.0 (www.spannend-projekt.de) funded by the Federal Company for Radioactive Waste Disposal (Bundesgesellschaft für Endlagerung, BGE). We thank two anonymous reviewers for their helpful comments on this manuscript.

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
