# Peer review of "A unified 3D geological model for Germany and adjacent areas"

_Earth System Science Data, 2025_

## Referee Comment (RC1)

**1 General comments**

In terms of significance, the data set presented and the corresponding article can be rated "1" (excellent) because (i) I regard the data as useful for future investigations of the thermal field, the mechanical state and the thermomechanical behavior of the crust all across Germany; (ii) the presented model cannot be built on a routine basis but relies on the efficiency of the new method; (iii) the data set is complete in the sense that a much larger number of input models has been combined than in previous models and the entire crust all over Germany is covered.

To improve the quality of the final model – now rated "2" – a more detailed description of the criteria (i) for choosing the 27 input models (while potentially disregarding alternatives) and (ii) for giving different priorities to spatially overlapping input models is required.

Concerning the presentation quality, the contribution can also be rated "2" since (i) the manuscript is clear and well structured, (ii) the figures additionally provided to illustrate all subsets of the data are very useful and efficient, and (iii) the manuscript in combination with the landing page for the dataset enables users to understand and (re-)use the data set. Some directions for improvements are given in the following, with specific comments provided separately for the manuscript (2.1) and for the data set (2.2).

**2 Specific comments**

**2.1 Manuscript**

The language of the manuscript is mostly consistent and precise, but an overall check should be done, e.g., (i) regarding comma placement, (ii) changing tenses when describing the work done (e.g., "we decided…", "we use…"), and (iii) simple typos, some of which but certainly not all are listed below.

The length of the article is appropriate. Its structure is clear except for some statements in the results chapter that should be moved to the discussion (see below). The figures and tables are clear and help to follow the lines of thoughts.

The method applied to build the model has not been used before and is well described. The approach of keeping a minimum thickness of 0.1 meters wherever a model unit is present is indeed similar to an approach used before (e.g., Anikiev et al. 2019), though in the latter case the minimum thickness of 0.1 meters for a unit was applied to the whole lateral extent of the model (including locations where a model unit is in fact not existing). Please add this to the discussion of advantages of the modelling approach.

**Abstract**

"[…]However, up to now only one attempt has been made to combine several of them to a Germany-wide model. We present a new Germany-wide 3D geological model […]" → Already here it should be stated why a new model is required (New input data and models available? Higher resolution required?).

**1 Introduction**

Line 19: 3D subsurface models are introduced as "showing lithostratigraphic horizons", which would restrict them to the domains above the crystalline basement. The presented model includes

crystalline crustal rocks as well. Their lateral juxtaposition would exclude them from a horizon / layer type model. Use "geological units" instead.

Line 22: The list of 3D models referred to seems to be arbitrary. What was the criterion for this choice of 3D models? If they have something particular in common, it should be stated.

Line 23: Suggestion: use "natural" instead of "undisturbed". Temperatures can be regarded as "disturbed" by natural processes as groundwater flow.

Line 39: "In the following we first present […]" → "In the following, we first shortly introduce […]"

**2 Model set up**

**2.1 Data base**
Line 48: It remains unclear which criteria have been applied to choose the 27 models used for model building. Please explain and potentially comment on models that have been disregarded.

Line 54: better "…depending on the position and the corresponding input model at place".

**2.2 Model correlation**
This chapter raises an important question: how do the authors deal with inconsistent input information from the different models used? For example, how to deal with the same stratigraphic interface shown at different depth levels? As this point is described in the following chapter (2.3), but the question comes up here already, the reader should at least be referred to the later description.

**2.3 Point set approach**
Line 106: "[…] which is tolerable for the model scale and resolution. The major advantages of the point set model approach […]" These parts are actually evaluative statements which should be moved to and deepened in the chapter "4. Discussion".

Line 100-111: "The order of projection is determined according to various criteria, e.g., year of publication, model resolution, etc." → This is an important point worth to be described in more detail. Please describe the philosophy behind this approach in more detail. The example given is not enough for transferring this modelling approach to another region, respectively another modelling campaign. For example, does the criterion "year of publication" mean that younger models are always given higher priority than older models, even if an older one covers a larger area or gives a higher resolution? To clarify this, a figure illustrating the decision-making process as a workflow would be helpful. Then the method can be re-used and referred to more easily.

Line 122: "The biggest advantage of the point set model, however, […]" → Again, this is part of an evaluation of the method and should be moved to the discussion.

In general, this chapter does not provide a detailed description of the computer programs developed to create an initial point set and multiply it several times for projecting it onto input data of various formats. How did the authors deal with input data of different formats?

We agree with the authors that an important advantage of the "point set approach" is that it can be used directly for the fast creation of discretized models with ApplePY (Ahlers et al. 2022; Ziegler et al. 2019) while no specific software is necessary. This should be emphasized more in the Discussion chapter.

**3 Results**

**3.1 Stratigraphic correlation**
[no comments]

**3.2 Model units**
[no comments]

**3.3 Presentation of results**
Line 164: Please clarify what your definition of "high stratigraphic resolution models" is?

**3.4 Generation of a discretized model**
Line 174-175: Please refer to a more detailed description of the format of an Abaqus *.inp file or provide the description here (as this would help re-using the approach).

**3.4.1 Worked example**
Line 187: "[…] whereby the element thickness increases with depth." → Please explain why and – if this helps to understand the reason – also how the increase of element thickness was designed.

Fig. 6, caption: For readers that are new to the field, it could be interesting to learn which software has been used for the visualization of the model.

**4. Discussion**
Line 232: "manageability of the final model" → Please clarify in which case(s) a model might not be manageable and in terms of what (computational times for the model production, visualization, …?).

Line 237: "Overall, the loss of information and vertical uncertainties are acceptable, especially considering the size and purpose of the model." → This statement is too general and thus requires some reasoning or at least a restriction to the applications intended by the authors.

Lines 248-271: These paragraphs include descriptions of results with some explanations of outcomes that could raise questions on the user's side. It's a list of examples the completeness level of which cannot be estimated by the reader. I suggest moving both paragraphs to chapter "3.3 Presentation of results".

In general, the discussion chapter should be used to deepen some thoughts on (i) the advantages and disadvantages of the modelling approach and the (ii) possibilities and limitations of using the new model, in particular by comparison to some already existing ones. On the methodological side, which problems could be expected if one wants to generate a triangulated mesh from the original model to better represent units that taper out? In terms of the usage of the geological model envisaged by the authors (i.e., geomechanical-numerical modelling), which changes and possibly improvements in the results can be expected compared to previous models (Ahlers et al. 2021, 2022a)?

**5. Conclusions**
The contents of this chapter actually is a summary of the manuscript and corresponding data set rather than a conclusion (the latter generally reflecting an opinion or a decision for action after considering all given information about something).

*Data availability*

It would be a great service to the community to share not only the final model but also the scripts, respectively applications, for creating a point set, multiplying it and projecting it onto input data of various formats.

*Supplement*

Supplementary material is provided as three tables (folder "essd-2025-320-supplement"). The 27 individual models used for building the new Germany 3D model are listed with their references in the file 2025_05_28_Table_S3.docx.

**2.2 Data set**

The data set is accessible via the given weblink (https://tudatalib.ulb.tu-darmstadt.de/handle/tudatalib/4615).

The data set is complete. The downloadable folder (Ahlers_2025_surfaces) contains 147 *.xyz-files according to the number of interfaces that define the 146 model units. Files can be opened with standard text editors without any problems. It should be added in the "Beschreibung" that "negative Z-values are in meters below sea level".

The downloadable folder "Ahlers_2025_units_figures" contains very useful figures for each of the model 146 units, providing easily accessible information (*.jpg format) about the spatial extent, geometry and input data distribution. Files can be opened with standard graphic viewers without any problems. The additional 11 files of combined plots are useful as well as they represent major stratigraphic units that have been differentiated also in previously published 3D models. The readme.txt file informs about the individual model units (their id's) used to build the combined plots.

The error of interface depths resulting from implementing a minimum layer thickness of 0.1 m is shortly discussed in the article (4. Discussion). Otherwise, there is nor quantification of the model uncertainties done.

The final structural model of Germany and adjacent areas is new in terms of its large extent combined with a relatively high geological and stratigraphic resolution. The model will certainly be useful in the future as a basis for running process simulations on various scales.

**Technical corrections**

Line 57:  → limitations

Line 58:  → for many types of

Line 65: Germany-wide

Line 132: either "stratigraphically independent" or "stratigraphy independent"

Line 162: indicates

Lines 184: the Netherlands

Line 268: shows

---

## Referee Comment (RC2)

**General comments**

I think this dataset is well worth publishing. It is relevant to society in general and a multitude of geological questions that can only be answered with such a unified model. The dataset represents a big step forward, as up to now, only small regional projects have been realised, or large-scale global models, but not with this scale of detail.

I have general comment about which model unit is included when models clash. In Line 110, onwards, – "the order of projection is determined according to various criteria, e.g. year of publication, model resolution, etc." This very vague statement seems to suggest the newest datasets are the best. I guess you must have decided on which stratigraphic layers were better for your end model using other criteria (a kind of triage!). Please expand this section. I realise you cannot discuss every nook and cranny of the model, but give some examples and give the rationale for your decision.

The weak point about this manuscript is the English language. Besides numerous typos and grammatical points, there are numerous errors concerning adverbs, verb use and use of the passive voice that obliterates from knowing whether the authors or other authors actually carried out the work. The use of capital letters, for instance for stratigraphic units, etc. needs to be revised and made uniform. Commas and hyphens are missing throughout the text. They would make the paper so much easier to read. Conversely, some commas are superfluous. The text should be checked by a native English speaker.  Don't use an apostrophe in "id's" -> ids (for instance in Figure 4 caption)

**Specific comments**

**Title**
I think the title is too short. What about "A combined geological model of Germany and adjacent areas". At least "for" should be replaced by "of".

**Abstract**
Line 11: We present a new Germany –wide 3D geological model -> We present such a 3D geological model

Line 13: change states -> countries

Line 15: was chosen which -> was chosen, which

Line 15: with regard to the flexibility -> with regards to its flexibility

Line 16: Write out "FE model" -> Finite-element model

**Introduction**
Line 19:  "lithostratigraphic" is too loose a term. These are all geological units, some which are lithostratigraphic, some are only model-based, e.g. the Moho.

Line 24:  "and how these conditions are potentially be disturbed by subsurface operations"
-> and how these conditions would be potentially disturbed by subsurface operations

Line 33: You give the abbreviations of the different models without introducing them to the reader. Either don't list the projects, just give the references, or write the acronyms out here.

Line 36: delete "currently available"

Line 43: finite element -> finite-element

Line 48: size -> sizes

Line 57: one model -> one single model,

Line 60: rephrase this sentence. "have been created" especially doesn't make sense.

Line 64: German-wide -> Germany-wide

Line 96: Rephase "we use point sets which are projected onto input data." I don't understand exactly what you mean here.

Line 119: "region models" – regional models

Line 142: delete "again"

Line 183: This sentence is passive. Please use the active, ie. "We chose a region…."

Line 185: "coordinates"

Line 189: "crystalline crust"

Line 199: "which have to be updated with respect" – There must be a better way to write this. I suspect modified and saved.

Figure 6: Can you please add some cultural information as an overlay? E.g. Rivers, political boundaries, etc.

Line 241. "A major advantage is that the point sets can be directly used for the fast creation of discretized models with ApplePY … as shown in Sect. 3.4 and no specific software is necessary for use."

-> A major advantage is that the point sets can be directly used to create discretized models with ApplePY … , as shown in Sect. 3.4. There is no need to use (other) specific software."

Line 245: in *to* one model.

Line 246: has some limitations, e.g.

Line 262: 3DD?

Line 263: was extended using – using what? If you mean the citations, bring the names out of the brackets.

Line 266: found in fragments -> found as fragments

Line274: Why not name all Germany's neighbouring countries? There are not that many.

---

## Author Comment (AC1)

**General comments**

In terms of significance, the data set presented and the corresponding article can be rated "1" (excellent) because (i) I regard the data as useful for future investigations of the thermal field, the mechanical state and the thermomechanical behavior of the crust all across Germany; (ii) the presented model cannot be built on a routine basis but relies on the efficiency of the new method; (iii) the data set is complete in the sense that a much larger number of input models has been combined than in previous models and the entire crust all over Germany is covered.

To improve the quality of the final model – now rated "2" – a more detailed description of the criteria (i) for choosing the 27 input models (while potentially disregarding alternatives) and (ii) for giving different priorities to spatially overlapping input models is required.

Concerning the presentation quality, the contribution can also be rated "2" since (i) the manuscript is clear and well structured, (ii) the figures additionally provided to illustrate all subsets of the data are very useful and efficient, and (iii) the manuscript in combination with the landing page for the dataset enables users to understand and (re-)use the data set. Some directions for improvements are given in the following, with specific comments provided separately for the manuscript (2.1) and for the data set (2.2).

Thank you very much for your review. Your suggestions and corrections helped improve this manuscript. We have added a comprehensive discussion of the criteria for selecting the 27 input models and the prioritization of overlapping models. We have addressed your specific comments below.

**Specific comments**

Manuscript

The language of the manuscript is mostly consistent and precise, but an overall check should be done, e.g., (i) regarding comma placement, (ii) changing tenses when describing the work done (e.g., "we decided…", "we use…"), and (iii) simple typos, some of which but certainly not all are listed below.

The length of the article is appropriate. Its structure is clear except for some statements in the results chapter that should be moved to the discussion (see below). The figures and tables are clear and help to follow the lines of thoughts.

The method applied to build the model has not been used before and is well described. The approach of keeping a minimum thickness of 0.1 meters wherever a model unit is present is indeed similar to an approach used before (e.g., Anikiev et al. 2019), though in the latter case the minimum thickness of 0.1 meters for a unit was applied to the whole lateral extent of the model (including locations where a model unit is in fact not existing). Please add this to the discussion of advantages of the modelling approach.

A minimum thickness of 0.1 m is also applied in our model to the whole lateral extent of the model, including locations where a model unit is in fact not existing. We have clarified this in lines 97-99.

*Abstract*

"[…]However, up to now only one attempt has been made to combine several of them to a Germany-wide model. We present a new Germany-wide 3D geological model […]" -> Already here it should be stated why a new model is required (New input data and models available? Higher resolution required?).

We have added a short explanation in lines 10-11: "Up to now only one attempt has been made to combine several of them to a Germany-wide model. However, there are many new models that have not been integrated into this model. Therefore, we present a new Germany-wide …".

*1 Introduction*

Line 19: 3D subsurface models are introduced as "showing lithostratigraphic horizons", which would restrict them to the domains above the crystalline basement. The presented model includes crystalline crustal rocks as well. Their lateral juxtaposition would exclude them from a horizon / layer type model. Use "geological units" instead.

You're right, we have replaced it with "geological units".

Line 22: The list of 3D models referred to seems to be arbitrary. What was the criterion for this choice of 3D models? If they have something particular in common, it should be stated.

They have nothing particular in common, however, we have chosen recent, thermal, hydraulic and geomechanical models within our study region, with a model size for which our model resolution could be of interest. As this list is only intended to provide a few examples and is not exhaustive, the selection is somewhat arbitrary.

Line 23: Suggestion: use "natural" instead of "undisturbed". Temperatures can be regarded as "disturbed" by natural processes as groundwater flow.

Thank you for pointing this out, we have replaced it.

Line 39: "In the following we first present […]" -> "In the following, we first shortly introduce […]"

*Thank you for your suggestion, we have changed it to* "In the following, we will briefly introduce"

*2 Model set up*

*2.1 Data base*

Line 48: It remains unclear which criteria have been applied to choose the 27 models used for model building. Please explain and potentially comment on models that have been disregarded.

Please see our answer below

Line 54: better "…depending on the position and the corresponding input model at place".

Thank you, we have added it.

*2.2 Model correlation*

This chapter raises an important question: how do the authors deal with inconsistent input information from the different models used? For example, how to deal with the same stratigraphic interface shown at different depth levels? As this point is described in the following chapter (2.3), but the question comes up here already, the reader should at least be referred to the later description.

Thank you very much for your suggestion, however, we think it is not necessary to refer to chapter 2.3 since it follows directly after.

*2.3 Point set approach*

Line 106: "[...] which is tolerable for the model scale and resolution. The major advantages of the point set model approach [...]" These parts are actually evaluative statements which should be moved to and deepened in the chapter "4. Discussion".

You're right, we have removed this sentence to the first paragraph in the discussion.

Line 100-111: "The order of projection is determined according to various criteria, e.g., year of publication, model resolution, etc." -> This is an important point worth to be described in more detail. Please describe the philosophy behind this approach in more detail. The example given is not enough for transferring this modelling approach to another region, respectively another modelling campaign. For example, does the criterion "year of publication" mean that younger models are always given higher priority than older models, even if an older one covers a larger area or gives a higher resolution? To clarify this, a figure illustrating the decision-making process as a workflow would be helpful. Then the method can be re-used and referred to more easily.

You are right that the criterion 'year of publication' is not precise enough and should not be generalized. Even if it could be applied appropriately in the German part of the Molasse basin. We have replaced the term with "year of model creation" and added a paragraph (lines 260-292) to the discussion that describes the evaluation criteria in greater detail.

However, a general workflow is very difficult to define, as criteria and available data vary depending on the model and region. However, the added detailed description of the assessment process within the German part of the Alpine Molasse Basin will hopefully help to improve the understanding.

Line 122: "The biggest advantage of the point set model, however, [...]" -> Again, this is part of an evaluation of the method and should be moved to the discussion.

You're right. We have removed this part.

In general, this chapter does not provide a detailed description of the computer programs developed to create an initial point set and multiply it several times for projecting it onto input data of various formats. How did the authors deal with input data of different formats?

We have used SKUA-GOCAD to create our unified model. We have added this information in line 114. All models were available as GOCAD files or xyz-files.

We agree with the authors that an important advantage of the "point set approach" is that it can be used directly for the fast creation of discretized models with ApplePY (Ahlers et al. 2022; Ziegler et al. 2019) while no specific software is necessary. This should be emphasized more in the Discussion chapter.

We have emphasized it in lines 253-254

*3 Results*

*3.1 Stratigraphic correlation*

[no comments]

*3.2 Model units*

[no comments]

*3.3 Presentation of results*

Line 164: Please clarify what your definition of "high stratigraphic resolution models" is?

We have added a reference to Fig.1.

*3.4 Generation of a discretized model*

Line 174-175: Please refer to a more detailed description of the format of an Abaqus *.inp file or provide the description here (as this would help re-using the approach).

A detailed description of the format of an Abaqus *.inp file is given by Ziegler et al. (2020a). We have added this information in lines 187-188.

*3.4.1 Worked example*

Line 187: "[...] whereby the element thickness increases with depth." ⬜ Please explain why and – if this helps to understand the reason – also how the increase of element thickness was designed.

This is a standard procedure because the reliability of geological information typically decreases with depth, which allows the number of elements to be reduced. We have added a reference to Ahlers et al. (2022a) in line 198.

Fig. 6, caption: For readers that are new to the field, it could be interesting to learn which software has been used for the visualization of the model.

We have added the software (Tecplot) to the figure caption.

*4. Discussion*

Line 232: "manageability of the final model" -> Please clarify in which case(s) a model might not be manageable and in terms of what (computational times for the model production, visualization, …?).

Manageability, particularly in terms of computing time and main memory. We have added this in line 242.

Line 237: "Overall, the loss of information and vertical uncertainties are acceptable, especially considering the size and purpose of the model." -> This statement is too general and thus requires some reasoning or at least a restriction to the applications intended by the authors.

We have added an application example in lines 247-250.

Lines 248-271: These paragraphs include descriptions of results with some explanations of outcomes that could raise questions on the user's side. It's a list of examples the completeness level of which cannot be estimated by the reader. I suggest moving both paragraphs to chapter "3.3 Presentation of results".

You're right, we have moved these paragraphs to chapter 3.3.

In general, the discussion chapter should be used to deepen some thoughts on (i) the advantages and disadvantages of the modelling approach and the (ii) possibilities and

limitations of using the new model, in particular by comparison to some already existing ones. On the methodological side, which problems could be expected if one wants to generate a triangulated mesh from the original model to better represent units that taper out? In terms of the usage of the geological model envisaged by the authors (i.e., geomechanical-numerical modelling), which changes and possibly improvements in the results can be expected compared to previous models (Ahlers et al. 2021, 2022a)?

We discuss the advantages and disadvantages of our model several times, particularly in the first two paragraphs. Chapter 3.4 provides a comprehensive option for further use. The advantages compared to the only existing Germany-wide model are the integration of new data sets (lines 32–36). The disadvantages compared to existing models result from the main disadvantage of a large-scale model, which is its resolution. This issue is addressed, e.g., in lines 242–247. In general, our model cannot replace or integrate all the details of smaller-scale models (e.g., those on a federal state scale), e.g., mentioned in lines 334–336.

The purpose for which the geological model is used strongly affects possible improvements or changes. One advantage for a geomechanical model is that stresses depend heavily on the Young's modulus of each unit; therefore, an improved geological model can enable better prediction. However, many other parameters, such as parameterization, also play a significant role. In our opinion, explaining these advantages and disadvantages in detail would exceed the scope of this publication.

We apologize, but we are uncertain if we have correctly understood your question: "On the methodological side, which problems could be expected if one wants to generate a triangulated mesh from the original model to better represent units that taper out?" Since point data sets are typical raw data for creating triangulated surfaces, no problems should arise.

*5. Conclusions*

The contents of this chapter actually is a summary of the manuscript and corresponding data set rather than a conclusion (the latter generally reflecting an opinion or a decision for action after considering all given information about something).

We have added a paragraph with possible improvements at the end of the conclusion chapter

*Data availability*

It would be a great service to the community to share not only the final model but also the scripts, respectively applications, for creating a point set, multiplying it and projecting it onto input data of various formats.

We have used SKUA-GOCAD for the model set-up. We have added this information in line 114

*Supplement*

Supplementary material is provided as three tables (folder "essd-2025-320-supplement"). The 27 individual models used for building the new Germany 3D model are listed with their references in the file 2025_05_28_Table_S3.docx.

2.2 Data set

The data set is accessible via the given weblink (https://tudatalib.ulb.tu-darmstadt.de/handle/tudatalib/4615).

The data set is complete. The downloadable folder (Ahlers_2025_surfaces) contains 147 *.xyz-files according to the number of interfaces that define the 146 model units. Files can be opened with standard text editors without any problems. It should be added in the "Beschreibung" that "negative Z-values are in meters below sea level".

Thank you very much for this suggestion, we have added "negative Z-values are in meters below sea level" to the description and the readme-file.

The downloadable folder "Ahlers_2025_units_figures" contains very useful figures for each of the model 146 units, providing easily accessible information (*.jpg format) about the spatial extent, geometry and input data distribution. Files can be opened with standard graphic viewers without any problems. The additional 11 files of combined plots are useful as well as they represent major stratigraphic units that have been differentiated also in previously published 3D models. The readme.txt file informs about the individual model units (their id's) used to build the combined plots.

The error of interface depths resulting from implementing a minimum layer thickness of 0.1 m is shortly discussed in the article (4. Discussion). Otherwise, there is nor quantification of the model uncertainties done.

The final structural model of Germany and adjacent areas is new in terms of its large extent combined with a relatively high geological and stratigraphic resolution. The model will certainly be useful in the future as a basis for running process simulations on various scales.

Thank you very much for these corrections!

Technical corrections

Line 57:  -> limitations

Line 58:  -> for many types of

Line 65: Germany-wide

Line 132: either "stratigraphically independent" or "stratigraphy independent"

Line 162: indicates

Lines 184: the Netherlands

Line 268: shows

---

## Author Comment (AC2)

**General comments**

I think this dataset is well worth publishing. It is relevant to society in general and a multitude of geological questions that can only be answered with such a unified model. The dataset represents a big step forward, as up to now, only small regional projects have been realised, or large-scale global models, but not with this scale of detail.

I have general comment about which model unit is included when models clash. In Line 110, onwards, – "the order of projection is determined according to various criteria, e.g. year of publication, model resolution, etc." This very vague statement seems to suggest the newest datasets are the best. I guess you must have decided on which stratigraphic layers were better for your end model using other criteria (a kind of triage!). Please expand this section. I realise you cannot discuss every nook and cranny of the model, but give some examples and give the rationale for your decision.

The weak point about this manuscript is the English language. Besides numerous typos and grammatical points, there are numerous errors concerning adverbs, verb use and use of the passive voice that obliterates from knowing whether the authors or other authors actually carried out the work. The use of capital letters, for instance for stratigraphic units, etc. needs to be revised and made uniform. Commas and hyphens are missing throughout the text. They would make the paper so much easier to read. Conversely, some commas are superfluous. The text should be checked by a native English speaker. Don't use an apostrophe in "id's" -> ids (for instance in Figure 4 caption)

Thank you very much for your helpful comments, suggestions, and corrections. They have helped us improve the manuscript. We have either incorporated your specific comments or provided comments.

You are right that the criterion "year of publication" is not precise enough to be generalized, even if it might be applicable in the German part of the Molasse Basin. We have replaced the term with "year of model creation" and have added a discussion of the "order of projection" (lines 258–290).

Additionally, we have proofread the entire manuscript for typos and grammatical errors. We have reduced the use of the passive voice, making it clear who performed the work. We have also restructured numerous sentences to improve readability and revised the use of capital letters.

**Specific comments**

**Title**

I think the title is too short. What about "A combined geological model of Germany and adjacentareas". At least "for" should be replaced by "of".

Thank you for your suggestion. We have changed it to „A unified 3D geological model of Germany and adjacent areas"

**Abstract**

Line 11: We present a new Germany –wide 3D geological model -> We present such a 3D geological model

We have rephrased the previous sentence. From our point of view, it now makes sense to mention these points again.

Line 13: change states -> countries

Line 15: was chosen which -> was chosen, which

Line 15: with regard to the flexibility -> with regards to its flexibility

Line 16: Write out "FE model" -> Finite-element model

Introduction

Line 19: "lithostratigraphic" is too loose a term. These are all geological units, some which are lithostratigraphic, some are only model-based, e.g. the Moho.

We have changed it to „geological units"

Line 24: "and how these conditions are potentially be disturbed by subsurface operations" -> and how these conditions would be potentially disturbed by subsurface operations

We have rephrased the sentence to „as well as how subsurface operations would potentially disturb them."

Line 33: You give the abbreviations of the different models without introducing them to the reader. Either don't list the projects, just give the references, or write the acronyms out here.

We have replaced MOLA and CEBS. However, TUNB, GeoMol and GeORG are the official model/project names.

Line 36: delete "currently available"

Line 43: finite element -> finite-element

Line 48: size -> sizes

Database

Line 57: one model -> one single model,

Line 60: rephrase this sentence. "have been created" especially doesn't make sense.

Line 64: German-wide -> Germany-wide

Line 96: Rephase "we use point sets which are projected onto input data." I don't understand exactly what you mean here.

We have rewritten the sentence: „Instead of creating triangulated surfaces, we use a point set approach."

Line 119: "region models" – regional models

Line 142: delete "again"

Line 183: This sentence is passive. Please use the active, ie. "We chose a region...."

Line 185: "coordinates"

Line 189: "crystalline crust"

Line 199: "which have to be updated with respect" – There must be a better way to write this. I

suspect modified and saved.

We have removed the whole sentence.

Figure 6: Can you please add some cultural information as an overlay? E.g. Rivers, political boundaries, etc.

We have added political boundaries, coastlines and country names.

Line 241. "A major advantage is that the point sets can be directly used for the fast creation of discretized models with ApplePY ... as shown in Sect. 3.4 and no specific software is necessary for use."

-> A major advantage is that the point sets can be directly used to create discretized models with ApplePY ... , as shown in Sect. 3.4. There is no need to use (other) specific software."

Line 245: in to one model.

Line 246: has some limitations, e.g.

Line 262: 3DD?

This is the name of the model of Anikiev et al., (2019). We have added a reference.

Line 263: was extended using – using what? If you mean the citations, bring the names out of the brackets.

We have removed the brackets.

Line 266: found in fragments -> found as fragments

Line274: Why not name all Germany's neighbouring countries? There are not that many.

We have added the missing countries.

---

## Author Response (AR2)

Dear Kirsten Elger,

We revised the new sections added during the last round of revisions in light of the referee's comments. We double-checked the grammar and American versus British English using DeeplWrite. Thank you very much for the suggestion.

Kind regards,

Steffen Ahlers